# Horizon Strings as 3d Black Hole Microstates

Arjun Bagchi[1,2], Daniel Grumiller[3,4], and M.M. Sheikh-Jabbari[5]

**1** Indian Institute of Technology Kanpur, Kanpur 208016, India

**2** Centre de Physique Theorique, Ecole Polytechnique de Paris, 91128 Palaiseau Cedex, France

**3** Institute for Theoretical Physics, TU Wien
Wiedner Hauptstrasse 8-10, A-1040 Vienna, Austria, Europe

**4** Theoretical Sciences Visiting Program, Okinawa Institute of Science and Technology
Graduate University, Onna, 904-0495, Japan

**5** School of Physics, Institute for Research in Fundamental Sciences (IPM),
P.O.Box 19395-5531, Tehran, Iran

abagchi@iitk.ac.in, grumil@hep.itp.tuwien.ac.at, jabbari@theory.ipm.ac.ir

## Abstract

We propose that 3d black holes are an ensemble of tensionless null string states. These microstates typically have non-zero winding. We evaluate their partition function in the limit of large excitation numbers and show that their combinatorics reproduces the Bekenstein–Hawking entropy and its semiclassical logarithmic corrections.

# 1  Introduction and motivation

Black holes and string theory have a long and intimate relationship, see [1–23] and Refs. therein. An emblematic result by Strominger and Vafa is the microscopic origin of the Bekenstein–Hawking (BH) entropy

$$S_{\text{BH}} = \frac{\text{Area}}{4G} \tag{1}$$

of certain extremal black holes [11]. The BH-law (1) is a template for falsification in quantum gravity, given the paucity of experimental data (see e.g. Sec. 10.2 in [24]). That is why the `hep-th` and `gr-qc` communities spent a lot of resources deriving the BH-law microscopically for more general black holes [25–36] (see also [37] for an alternative proposal).

Despite these efforts, it is unclear how to construct stringy microstates for non-extremal black holes, which is an obstacle to further progress in string theory and black holes since extremal black holes are zero-temperature states that do not occur in Nature. Another slightly puzzling aspect of (1) is its universality: For large black holes, the entropy depends only on the horizon area and not on any details of the microstates.

A complementary approach to deriving the BH-law (1) is to focus on near-horizon symmetries, see e.g. [38–56]. In these derivations, the universality of the BH-law (1) and its applicability to non-extremal black holes become more transparent, but the precise identification of the microstates typically remains obscure or requires ad-hoc input: Even when succeeding in counting the microstates from some horizon-symmetry arguments it remains unclear what these microstates are.

The goal of our paper is to combine both approaches and construct, based on horizon-symmetry considerations, stringy microstates of non-extremal black holes. For technical reasons, we restrict ourselves to three-dimensional (3d) black holes (BTZ black holes) [57].

We postulate that the string worldsheet relevant for the microscopic description of the BTZ black hole coincides with the black hole horizon, a null hypersurface generated by some null vector. The symmetries preserving such a null hypersurface are diffeomorphisms of the null hypersurface and scalings of the null vector [58, 59]. In 3d, these are precisely the symmetries of the worldsheet of a null string theory [60–66], thus motivating our postulate. Phrased more explicitly, we propose that a 3d black hole is an ensemble of null string states. The worldsheet of these strings may be identified with the horizon of the black hole they collectively represent.

Our main result derived from this postulate (and additional assumptions that we shall spell out) is an explicit list of BTZ microstates in terms of null string excitations (explained below) and proof that their number accounts for the correct BH-law (1) as well as the subleading semiclassical corrections thereof (see [67] and Refs. therein).

Before delving into details, we present some further intuition behind our formulation. We begin with the basic observation that the horizon of a generic black hole is a co-dimension one null surface. We aim to replace the black hole with some closed string with a null worldsheet

associated with this null surface. The quantization of this null string yields a collection of null string states, which are supposed to be the microstates of the black hole.

There are a couple of important ways that the null string is specific to the black hole it describes. The first point concerns symmetries. In any effective description of a physical system, it is crucial to preserve the appropriate set of symmetries. We shall demonstrate that the near-horizon symmetries associated with null surfaces (such as black hole horizons) coincide with the symmetries of the worldsheet action of tensionless null strings. The second point is that the string generically winds around a direction in the ambient 3d spacetime. The radius of the compact direction naturally is identified with the main macroscopic scale in the system, the radius of the event horizon.

Our proposal entails that a black hole of a certain mass $M$ is a coarse-grained object that arises as an effective description of certain high-excitation sectors of this fundamental null string. The vibrational modes of this string at a high enough level $N$ yield the microstates of this coarse-grained structure. Inspired by results from near-horizon physics, in particular, a matching of the near-horizon and null string symmetries and the near-horizon first law (on which we shall elaborate later), we relate the mass $M$ and the level $N$ in a specific way. This is the third point of contact between the null string picture and the black hole picture, which then yields an explicit list of BTZ microstates, whose combinatorics reproduces the BH entropy (1) and its logarithmic corrections.

This paper is organized as follows. In Section 2, we recap tensionless null strings. In Section 3, we review the near-horizon expansion and symmetries and match the latter with the symmetries of tensionless null strings. In Section 4, based on tensionless null strings we define horizon strings and build the associated physical Hilbert space. In Section 5, we provide an explicit list of all BTZ black hole microstates, labeled by the horizon string quantum numbers: mode excitation numbers, winding number, and momentum number. Moreover, we consider the combinatorics of these microstates in the limit of large occupation numbers, corresponding to the large mass limit. In Section 6, we compare the results of our combinatorial analysis with the Bekenstein–Hawking law. In Section 7, we conclude with a discussion and an outlook.

## 2   Recap of tensionless null strings

The study of null strings was initiated by Schild in the 1970s [60], corresponding to a limit where the worldsheet of the string becomes a null surface. Analogous to the massless limit of a point particle whose worldline is null, a null string limit necessitates sending the tension of the string to zero [62]. These tensionless null strings are the focus of this paper. The null string thus explores the limit of string theory where the only parameter in the free theory, $\alpha'$, is sent to infinity and hence is the opposite of the point particle limit, $\alpha' \to 0$, where the dynamics of a string is essentially replaced by its center of mass dynamics. In the tensionless limit, the fundamental string is floppy and can become macroscopically long.

### 2.1   Classical tensionless null strings

Our formulation of the tensionless null string follows [61], starting with the well-known Polyakov action of the tensile string,

$$S = -\frac{T}{2} \int d\tau \, d\sigma \, \sqrt{\gamma} \, \gamma^{ab} \partial_a X^\mu \partial_b X^\nu \, G_{\mu\nu}(X) \tag{2}$$

where $T \propto 1/\alpha'$ is the string tension, $\gamma^{ab}$ is the worldsheet metric, and $G_{\mu\nu}$ is the metric on which the string propagates. In the tensionless null string limit, the worldsheet metric degenerates, and

the equivalent of the Polyakov action becomes [61]

$$S = \frac{\kappa}{2} \int d\tau \, d\sigma \, V^a V^b \partial_a X^\mu \partial_b X^\nu \, \eta_{\mu\nu}(X) \,. \tag{3}$$

Here, $V^a$ are weight $\frac{1}{2}$ vector densities and $V^a V^b$ replaces $\sqrt{\gamma} \, \gamma^{ab}$ in the tensionless limit. In terms of recent terminology, the worldsheet of a null string is described by a 2d Carrollian geometry [68–70], see section 2.2 for more on the Carrollian picture. The constant $\kappa$ is put in to match dimensions, and we shall have more to say about it in our construction later. We have also chosen the spacetime metric to be flat in this review section.

The tensionless worldsheet, like its tensile counterpart, enjoys diffeomorphism invariance where the vector density $V^a$ transform as

$$\delta_\xi V^a = \xi^b \partial_b V^a - V^b \partial_b \xi^a + \frac{1}{2} \, V^a \, \partial_b \xi^b \,. \tag{4}$$

We thus need to fix the gauge. A particularly convenient one is the equivalent of the conformal gauge where we choose

$$V^a \partial_a = \partial_\tau \,. \tag{5}$$

Analogously to the tensile case, this choice does not completely fix the gauge and we are left with residual worldsheet diffeomorphisms generated by the vector fields [63]

$$\xi = (h(\sigma) + \tau f'(\sigma)) \, \partial_\tau + f(\sigma) \, \partial_\sigma \tag{6}$$

that preserve the gauge choice (5), $\delta_\xi V^a = 0$. Defining generators

$$L(f) = f'(\sigma) \tau \, \partial_\tau + f(\sigma) \, \partial_\sigma \qquad\qquad M(g) = g(\sigma) \, \partial_\tau \tag{7}$$

and expanding in Fourier modes, we find the residual symmetry algebra on the tensionless worldsheet

$$[L_n, L_m] = (n - m) \, L_{n+m} \qquad [L_n, M_m] = (n - m) \, M_{n+m} \qquad [M_n, M_m] = 0 \,. \tag{8}$$

This is the Bondi–van der Burgh–Metzner–Sachs algebra in 3d (BMS$_3$), see e.g. [64, 71–73]. The BMS algebra was earlier found in the context of asymptotic symmetries of flat spacetime at its null boundary [73–75] and has been of relevance for attempts to construct a holographic correspondence in flat spacetimes [46,75–78]. The recent revival of the study of tensionless strings is based on the fact that this BMS algebra replaces the two copies of Virasoro algebra on the worldsheet of the tensionless string. The (centerless) BMS algebra (8) is central to the organization of the tensionless string.

In the gauge (5), the equations of motion of the tensionless string simplify to

$$\ddot{X} = 0 \tag{9}$$

and the constraints read

$$\dot{X}^2 = 0 = \dot{X} \cdot X' = 0 \,. \tag{10}$$

The equations of motion are solved by the mode expansion [63]

$$X^\mu(\tau, \sigma) = x^\mu + A_0^\mu \sigma + B_0^\mu \tau + i \sum_{n \neq 0} \frac{1}{n} \left( A_n^\mu - in\tau B_n^\mu \right) e^{-in\sigma} \,. \tag{11}$$

We are interested in studying closed strings and hence demand the boundary condition

$$X^\mu(\tau, \sigma) = X^\mu(\tau, \sigma + 2\pi) \,. \tag{12}$$

This renders $A_0 = 0$. Plugging the mode expansion into the constraints yields

$$\dot{X} \cdot X' = \frac{1}{2} \sum_{n,m} (A_{-m} - in\tau B_{-m}) B_{n+m} e^{-in\sigma} = \sum_n (L_n - in\tau M_n) e^{-in\sigma} = T_1 = 0 \tag{13a}$$

$$\dot{X}^2 = \frac{1}{2} \sum_{n,m} B_{-n} B_{n+m} e^{-in\sigma} = \sum_n M_n e^{-in\sigma} = T_2 = 0. \tag{13b}$$

Above, we used the definitions

$$L_n = \frac{1}{2} \sum_m A^\mu_{-m} B^\nu_{n+m} \eta_{\mu\nu} \qquad\qquad M_n = \frac{1}{2} \sum_m B^\mu_{-m} B^\nu_{n+m} \eta_{\mu\nu} \tag{14}$$

of the BMS generators. The quantities $T_1, T_2$ are two components of the energy-momentum tensor for 2d BMS$_3$-invariant field theories see e.g. [79–83] and references therein. The constraint equations translate to the vanishing of the energy-momentum tensor of the worldsheet theory,

$$T_1 = 0 = T_2. \tag{15}$$

The existence of these constraints and the possibility of winding (12) are key differences between null strings and a collection of free particles.

The algebra of the modes $L, M$ yields the BMS algebra (8) provided the non-vanishing Poisson brackets of the $A, B$ modes are given by

$$\{A^\mu_n, B^\nu_m\} = -2in\, \delta_{n,m}\, \eta^{\mu\nu}. \tag{16}$$

The algebra (16) is not that of harmonic oscillators. To switch to a harmonic oscillator basis, we change the basis,

$$C^\mu_n = 2 \left( A^\mu_n + B^\mu_n \right) \qquad\qquad \tilde{C}^\mu_n = 2 \left( -A^\mu_{-n} + B^\mu_{-n} \right). \tag{17}$$

The Poisson brackets between the $C$ and $\tilde{C}$,

$$\{C^\mu_n, C^\nu_m\} = -in\, \delta_{n,m}\, \eta^{\mu\nu} \qquad \{\tilde{C}^\mu_n, \tilde{C}^\nu_m\} = -in\, \delta_{n,m}\, \eta^{\mu\nu} \qquad \{C^\mu_n, \tilde{C}^\nu_m\} = 0 \tag{18}$$

are those of harmonic oscillators.

## 2.2   Tensionless limit as a worldsheet Carroll limit

In the tensionless limit, the string becomes long and floppy. This can be achieved in terms of worldsheet coordinates $(\sigma, \tau)$ by the singular scaling

$$\tau \to \epsilon\tau \qquad\qquad \sigma \to \sigma \qquad\qquad \epsilon \to 0. \tag{19}$$

Intuitively, the spatial direction becomes large on the worldsheet, typifying a long tensionless string. The scaling (19) is a Carrollian limit on the worldsheet, where the worldsheet speed of light goes to zero. The emergence of a Carrollian structure on the worldsheet is a sign that the tensionless worldsheet becomes null [63].

The limit (19) leads to a contraction of two copies of the Virasoro algebra,

$$\mathcal{L}_n - \bar{\mathcal{L}}_{-n} = L_n \qquad\qquad \mathcal{L}_n + \bar{\mathcal{L}}_{-n} = \frac{1}{\epsilon} M_n \tag{20}$$

to the 2d Carroll algebra, which is isomorphic to the BMS$_3$ algebra [46, 84] that arises on the worldsheet (8).

The analysis of the mode expansion of the equation of motion and the constraints presented in the previous section can also be obtained by implementing (19) on the tensile string mode expansions [63, 64]

$$\tilde{X}^\mu(\sigma, \tau) = x^\mu + 4\sqrt{\frac{\alpha'}{2}}\,\alpha_0^\mu + i\sqrt{\frac{\alpha'}{2}}\left(\alpha_n^\mu e^{-in(\tau+\sigma)} + \tilde{\alpha}_n^\mu e^{-in(\tau-\sigma)}\right) \tag{21}$$

where in addition we need to scale $\alpha' \to \kappa/\epsilon$. This leads to a relation between the tensile oscillators $(\alpha, \tilde{\alpha})$ and the tensionless ones $(A, B)$,

$$A_n = \frac{1}{\sqrt{\epsilon}}(\alpha_n - \tilde{\alpha}_{-n}) \qquad B_n = \sqrt{\epsilon}(\alpha_n + \tilde{\alpha}_{-n})\,, \tag{22}$$

where we have suppressed the target space indices $\mu, \nu$ for the ease of notation. Plugging these expressions into the relation of the oscillator modes with the Virasoro algebra, $\mathcal{L}_n = \sum_n \alpha_{n+m}\alpha_{-m}$, and the ensuing BMS version (14), recovers (20), thus providing us with a sanity check of our various formulae.

Finally, we note that the relation between the tensile $\alpha$ oscillators and the tensionless $C$ oscillators is given by the Bogoliubov transformation

$$C_n = \beta_+\alpha_n + \beta_-\tilde{\alpha}_{-n} \qquad \tilde{C}_n = \beta_-\alpha_{-n} + \beta_+\tilde{\alpha}_n \tag{23}$$

where $\beta_\pm = \sqrt{\epsilon} \pm \frac{1}{\sqrt{\epsilon}}$. Despite the singularity of the tensionless limit, the Poisson structure in terms of these sets of oscillators is preserved as $\epsilon$ tends to zero.

## 2.3 Quantizing the null string

Having reviewed the salient features of the classical tensionless string, we now turn our attention to the quantization of these objects. The discussion here will closely mirror that of [64,85]. The usual tensile string is quantized as a free 2d relativistic massless scalar field. The constraints are then imposed on the Hilbert space to give a space of physical states. Our construction of the tensionless quantum theory proceeds along similar lines, where we now quantize a free 2d Carrollian massless scalar field and impose the quantum versions of the classical constraints (15),

$$\langle\text{phys}'|T_1|\text{phys}\rangle = 0 = \langle\text{phys}'|T_2|\text{phys}\rangle \tag{24}$$

where $|\text{phys}\rangle, |\text{phys}'\rangle$ are any two physical states of the tensionless theory. In terms of the modes of the energy-momentum tensor (13a), (13b), these become

$$\langle\text{phys}'|L_n|\text{phys}\rangle = 0 = \langle\text{phys}'|M_n|\text{phys}\rangle\,. \tag{25}$$

In [64], it was shown that three different quantum mechanical systems emerged out of the classical null string that we discussed in the previous section. The gist of their canonical quantization is that for each set of oscillator $O_n = \{L_n, M_n\}$ there are three different ways of imposing the constraints, viz.,

1. $O_n|\text{phys}\rangle = 0 \quad \forall n > 0$.

2. $O_n|\text{phys}\rangle = 0$ for $n \neq 0$, and

3. $O_n|\text{phys}\rangle \neq 0$ but $\langle\text{phys}'|O_n|\text{phys}\rangle = 0$ for $n \neq 0$.

This is true for both sets of oscillators leading to a total of nine possibilities. Assuming the vacuum to be a physical state, i.e., $\langle 0|O_n|0\rangle = 0$ for $n \neq 0$, one can show that the closure of the BMS algebra eliminates many of the possibilities and we are left with three consistent choices of vacua.

- **Flipped Vacuum** $|0\rangle_F$: This is the vacuum where the BMS$_3$ algebra is imposed in terms of highest weight representations.

$$L_n|0\rangle_F = 0 \qquad\qquad M_n|0\rangle_F = 0 \qquad\qquad \forall n > 0\,. \qquad (26)$$

The theory built on this vacuum is the bosonic ambitwistor theory [86, 87] and has some strange features, including a restricted spectrum. Since the limit from the Virasoro generators of the parent tensile theory (20) mixes creation and annihilation operators, the parent of the Flipped theory is an unusual string theory where the vacuum is annihilated by the usual annihilation operators $\alpha_n$ on the right, but by the creation operators $\tilde{\alpha}_{-n}$ on the left ($n > 0$). This is the reason behind the nomenclature. We are not interested in this vacuum in the present work.

- **Induced Vacuum** $|0\rangle_I$: The condition for this vacuum is given by

$$M_n|0\rangle_I = 0,\ \forall n \neq 0 \qquad\qquad L_n|0\rangle_I \neq 0,\ \text{but}\ {}_I\langle 0|L_n|0\rangle_I = 0,\ \text{for}\ n \neq 0\,. \qquad (27)$$

The theory built on this vacuum falls under the so-called induced representation of the BMS$_3$ algebra, see [88]. This vacuum follows directly from the limit of the usual tensile theory (20), where the Virasoro highest weights give rise to the BMS$_3$ induced representations in the Carrollian limit. The theory obtained is thus also what one gets if one takes a straightforward high-energy limit on tensile bosonic string theory. We are not interested in this vacuum in the present work.

- **Oscillator Vacuum** $|0\rangle_O$: The condition for this vacuum is given by

$$L_n|0\rangle_O \neq 0 \qquad\quad M_n|0\rangle_O \neq 0 \qquad \text{but}\ {}_O\langle 0|L_n|0\rangle_O = {}_O\langle 0|M_n|0\rangle_O = 0,\ \text{for}\ n \neq 0\,. \qquad (28)$$

This is the vacuum of the $C$-oscillators described earlier in (17), viz.,

$$C_n^\mu|0\rangle_O = 0 \qquad\qquad \tilde{C}_n^\mu|0\rangle_O = 0 \quad \forall n \neq 0\,. \qquad (29)$$

This vacuum leads to an intriguing theory, which comes about naturally from an accelerated worldsheet theory [66, 85]. The null string worldsheet can be thought about as the end point of a series of accelerating worldsheets. The accelerated worldsheet is the 2d analog of an accelerated observer in 2d Rindler spacetime. The null worldsheet is the string equivalent of a Rindler observer hitting the Rindler horizon. The Bogoliubov transformations (23) are the singular limit of the transformations between the inertial Minkowski observer and the accelerating Rindler observer. We pick this vacuum for our construction of the BTZ black hole microstates for three a priori reasons: 1. The Rindler limit is physically meaningful when approaching a non-extremal black hole horizon, 2. The oscillator algebra (18) concurs with the near-horizon symmetries of the BTZ black hole reviewed in the next section, 3. The Oscillator Vacuum is the weakest of the three vacuum conditions and thus allows for the largest number of physical states. A posteriori, another reason for this vacuum choice is that it yields the correct combinatorics to match the BH entropy (1).

## 3   Null strings, black holes, and their symmetries

In this section, we remind the reader of the near-horizon expansion of the BTZ black hole and the associated symmetries. We then relate these symmetries to those of null strings, which provides the basis of our proposal to consider a theory of null strings as appropriate for a black hole description.

### 3.1 Near-horizon expansion of BTZ

The most general near-horizon expansion of non-extremal BTZ black holes is given in Eq. (5) of [52]. In a co-rotating frame and (ingoing) Eddington–Finkelstein coordinates, the full BTZ metric adapted to a near-horizon expansion,

$$ds^2 = -2a\rho f(\rho)\, d\hat{v}^2 + 2\, d\hat{v}\, d\rho + 4\omega\rho f(\rho)\, d\hat{v}\, d\phi - 2\frac{\omega}{a}\, d\phi\, d\rho + \left[R_h^2 + \frac{2\rho}{a\ell^2}(R_h^2 - \ell^2\omega^2)f(\rho)\right] d\phi^2 \tag{30}$$

contains the function $f(\rho) = 1 + \rho/(2a\ell^2)$, the AdS radius $\ell$, surface gravity $a$, the near-horizon angular momentum $\omega$, and the horizon radius $R_h$. We assume for simplicity constant $\omega$ and make the coordinate shift $v = \hat{v} - \frac{\omega}{a}\phi$, yielding

$$ds^2 = -2a\rho f(\rho)\, dv^2 + 2\, dv\, d\rho + R_h^2\left(1 + \frac{2\rho}{a\ell^2}f(\rho)\right) d\phi^2\,. \tag{31}$$

Note that the near-horizon angular momentum $\omega$ has dropped out in (31). So this metric is suitable for ensembles of fixed angular momentum and either fixed or varying horizon radius.

Since we are interested in evaluating the metric (31) on the horizon, we do not care about local curvature effects and thus, to leading order, neglect effects from the AdS radius,

$$ds^2 \approx -2a\rho\, dv^2 + 2\, dv\, d\rho + R_h^2\, d\phi^2\,. \tag{32}$$

The metric (32) describes a geometry that is a direct product of 2-dimensional Rindler space (spanned by $v$ and $\rho$) and a round $S^1$ with radius $R_h$.

With the usual coordinate transformation $v = t + \frac{\ln\rho}{2a}$ the metric (32) is converted into Schwarzschild-gauge

$$ds^2 \approx -2a\rho\, dt^2 + \frac{d\rho^2}{2a\rho} + R_h^2\, d\phi^2\,. \tag{33}$$

The additional change of radial coordinate $r = \sqrt{2\rho/a}$ brings the metric into Rindler gauge

$$ds^2 \approx -a^2 r^2\, dt^2 + dr^2 + R_h^2\, d\phi^2\,. \tag{34}$$

Finally, we introduce lightcone coordinates $x^{\pm} = \pm\frac{1}{\sqrt{2}}r\, e^{\pm at}$ to bring the Rindler metric into Minkowski form,

$$ds^2 \approx -2\, dx^+ dx^- + R_h^2\, d\phi^2 = G_{\mu\nu}\, dx^\mu\, dx^\nu\,. \tag{35}$$

This metric is the starting point for our analysis of null strings on the (non-extremal) BTZ black hole horizon, where it describes our target space geometry.

The non-trivial aspects of the target space metric (35) are the periodicity of the angular coordinate, $\phi \sim \phi + 2\pi$, and the appearance of a (macroscopic) scale, the horizon radius $R_h$. Thus, despite being a locally flat geometry, our target space metric differs from 3d Minkwoski spacetime in two aspects: 1. it has a compact direction $\phi$, and 2. it carries a physical scale $R_h$. Both aspects are crucial for our construction.

### 3.2 Null boundary and null worldsheet symmetries

We consider now symmetries as a guiding principle, starting with a general 3d metric in which $r = 0$ is a null surface,[1]

$$ds^2 = -rV\, dv^2 + 2\eta\, dv\, dr + \mathcal{R}^2\, (d\phi + U\, dv)^2\,. \tag{36}$$

---

[1]For technical reasons, the gauge used here differs slightly from the gauge used in (30); namely, the $d\rho\, d\phi$-term present there is fixed to be absent here. This gauge fixing has no dramatic consequences.

Here $V, \mathcal{R}, U$ are generic functions of all coordinates $r, v, \phi$, which we assume to be smooth around $r = 0$, and $\eta = \eta(v, \phi)$. The reader can think of the locus $r = 0$ as the black hole horizon, though it is not necessary to adopt this viewpoint.

The family of metrics (36) is preserved by the near-horizon diffeomorphisms [58]

$$\xi = T\partial_v - rW\partial_r + Y\partial_\phi + \cdots \tag{37}$$

where $\Omega = \mathcal{R}(r = 0)$, the ellipsis denotes subleading terms, and $T$, $W$, and $Y$ generic functions of $v, \phi$. Their Lie bracket algebra

$$[\xi(T_1, W_1, Y_1), \xi(T_2, W_2, Y_2)]_{\text{Lie}} = \xi(T_{12}, W_{12}, Y_{12}) \tag{38}$$

closes with the structure functions

$$T_{12} = (T_1\partial_v + Y_1\partial_\phi)T_2 - (1 \leftrightarrow 2) \tag{39a}$$
$$Y_{12} = (T_1\partial_v + Y_1\partial_\phi)Y_2 - (1 \leftrightarrow 2) \tag{39b}$$
$$W_{12} = (T_1\partial_v + Y_1\partial_\phi)W_2 - (1 \leftrightarrow 2). \tag{39c}$$

The functions $T, Y$ together generate $\text{diff}_2$, while $W$ generates radial dilatations and transforms like a scalar under $\text{diff}_2$. Since $r = 0$ is a null surface generated by $\partial_r$, the dilatation $W$ also scales the other null vector in the $rv$-plane, $\partial_v$.

A key observation is that the null boundary algebra (39) exactly matches the null string worldsheet symmetries. Indeed, the null string action (3) is also invariant under $\text{diff}_2$ and under dilatations (the latter invariance is a consequence of the conformal Carrollian structure on the worldsheet, see [62, 63] for details). The fact that the respective symmetries match is a non-trivial consistency check of our proposal.

## 3.3 Near-horizon symmetries and glimpse of Oscillator Vacuum

We have just performed a rather generic check of symmetries, finding a match between symmetries of null hypersurfaces and the worldsheet action of tensionless null strings.

Now we consider a refined set of symmetries, namely near-horizon symmetries associated with a BTZ black hole at fixed surface gravity. These symmetries act non-trivially on the near-horizon state space much in the same way that the Brown–Henneaux Virasoro symmetries act on the asymptotic state space, see [51] for details.

In the coordinates (30), the near-horizon boundary conditions require fixed $a$ but allow state- and angle-dependent fluctuations of the horizon radius $R_h$ and the near-horizon angular momentum $\omega$.[2] The near-horizon charges found in [51] are suitably normalized Fourier modes of these state-dependent functions

$$J_n \sim \oint \mathrm{d}\phi \, e^{in\phi} \left(R_h(\phi) + \omega(\phi)\right) \qquad \tilde{J}_n \sim \oint \mathrm{d}\phi \, e^{in\phi} \left(R_h(\phi) - \omega(\phi)\right). \tag{40}$$

Their algebra,

$$[J_n, J_m] = n\, \delta_{n+m, 0} = [\tilde{J}_n, \tilde{J}_m] \tag{41}$$

consists of two $\hat{u}(1)$ currents. The standard vacuum conditions

$$J_n|0\rangle = 0 = \tilde{J}_n|0\rangle \qquad n \in \mathbb{Z}^+ \tag{42}$$

led to soft Heisenberg hair when acting with descendants, $J_n, \tilde{J}_n$ on the vacuum state $|0\rangle$. The attribute "soft" physically refers to the zero energy of the descendants. Algebraically, softness

---

[2]Assuming constant $a$, these functions are $v$-independent as a consequence of the near-horizon holographic Ward identities, see [52].

comes from the fact that the near-horizon Hamiltonian, $H \sim J_0 + \tilde{J}_0$, commutes with all elements of the near-horizon symmetry algebra (41).

The symmetries (41) and vacuum conditions (42) are reminiscent of the Oscillator Vacuum of tensionless null strings. The next section is going to make this observation more precise.

# 4   Horizon strings

In this section, we define and construct horizon strings, based on tensionless null strings that wrap a spatial circle, with a suitable vacuum choice that matches with the near-horizon symmetries reviewed in the previous section.

## 4.1   Basic set-up of horizon strings

The strings we consider are described by (3) with the background target space metric $G_{\mu\nu}$ as in (35). For later convenience, we introduce $\varphi := R_h \phi$, where $\phi \sim \phi + 2\pi$ and $\varphi \sim \varphi + 2\pi R_h$, while the lightcone coordinates $x^\pm$ take arbitrary real values. The gauge-fixed version of the null string action (3) for the target space metric (35) simplifies to

$$\mathcal{S}_{\text{gf}} = \frac{\kappa}{2} \int d\tau \, d\sigma \left( -2(\partial_\tau X^+)(\partial_\tau X^-) + (\partial_\tau X^\varphi)^2 \right) \tag{43}$$

where $X^\varphi = R_h X^\phi$. Varying the gauge-fixed action (43) yields the equation of motion

$$\partial_\tau^2 X^\mu = 0 \tag{44}$$

solved as before by the mode expansion

$$X^\mu(\tau, \sigma) = x^\mu + A_0^\mu \sigma + B_0^\mu \tau + i \sum_{n \neq 0} \frac{1}{n} \left( A_n^\mu - in\tau B_n^\mu \right) e^{-in\sigma} \,. \tag{45}$$

Additionally, we now allow closed strings to wind around the compact $\varphi$-direction,

$$X^\varphi(\sigma + 2\pi, \tau) = X^\varphi(\sigma, \tau) + 2\pi R_h \, \omega \qquad \omega \in \mathbb{Z} \tag{46}$$

and have identifications $X^\pm(\sigma + 2\pi, \tau) = X^\pm(\sigma, \tau)$, implying

$$A_0^\varphi = R_h \omega \qquad\qquad A_0^\pm = 0 \,. \tag{47}$$

As usual in string theory [89, 90], the momentum along the circle is quantized in units of one over its radius.

$$p^\varphi = \kappa B_0^\varphi = \frac{n}{R_h} \qquad n \in \mathbb{Z} \tag{48}$$

We refer to null strings on the background metric (35) that wind around the $\phi$-direction as "horizon strings".

## 4.2   Quantizing horizon strings

The quantization of horizon strings parallels that of a generic null string discussed in section 2.3, with the addition that horizon strings can wrap around the $\phi$ direction and have non-zero winding, which does affect the quantization, as we shall demonstrate.

The constraints are (10) or (15) in the classical theory and (24) in the quantum theory. The first step towards quantization of horizon strings is the choice of the appropriate vacuum state among the three possibilities discussed in section 2.3, guided by the near-horizon discussion in section 3.

### 4.2.1 Vacuum choice for horizon strings

In the lightcone gauge, only the $\varphi$-component matters because one can fix the residual diffeomorphisms such that $X^+ = x^+ + B_0^+ \tau$ and $X^-$ is determined by $X^\varphi$, see [91]. To quantize horizon strings, we choose the Oscillator Vacuum (42) for this remaining component, i.e., the operators

$$J_n := \sqrt{\tfrac{\kappa}{2}} \left( A_n^\varphi + B_n^\varphi \right) \qquad \tilde{J}_n := \sqrt{\tfrac{\kappa}{2}} \left( - A_{-n}^\varphi + B_{-n}^\varphi \right) \tag{49}$$

obey the oscillator algebra (41). This algebraic equivalence to the soft Heisenberg hair algebra provides confidence that the Oscillator Vacuum is appropriate for BTZ black holes.

### 4.2.2 Hilbert space

To construct the physical Hilbert space, we start with classifying states at different levels $N \in \mathbb{N}$,

$$\text{Vacuum: } |0, p^\mu, \omega\rangle \equiv |0\rangle$$
$$\text{Level 1: } J_{-1}|0\rangle, \tilde{J}_{-1}|0\rangle$$
$$\text{Level 2: } J_{-2}|0\rangle, J_{-1}^2|0\rangle, J_{-1}\tilde{J}_{-1}|0\rangle, \tilde{J}_{-1}^2|0\rangle, \tilde{J}_{-2}|0\rangle$$
$$\cdots$$

where $p^\mu = \kappa B_0^\mu$. The level splits into two integers associated with each set of oscillator modes, $N = r + s$. For instance, the five states above at level $N = 2$ have, respectively, $(r, s) = (2, 0)$, $(r, s) = (2, 0)$, $(r, s) = (1, 1)$, $(r, s) = (0, 2)$ and $(r, s) = (0, 2)$. The levels $r, s$, in turn, can be decomposed into a collection of integers associated with individual creation operators, $r = \sum_n n r_n$ and $s = \sum_n n s_n$. For example, the first two states at level $N = 2$ with $r = 2$ and $s = 0$ split, respectively, into $r_2 = 1$ and $r_1 = 2$ (with all other $r_i = 0$ in each case).

A generic state in the above set,

$$|\Psi\rangle = |p^\mu, \{r_i\}, \{s_i\}, \omega\rangle, \tag{50}$$

is given by arbitrary combinations of the creation operators $J_{-m}, \tilde{J}_{-m}$ acting on the Oscillator Vacuum $|0, p^\mu, \omega\rangle$. Physical states are a subclass of these generic states subject to the constraints (28). We are interested in physical states without momentum in the radial direction, which in our lightcone coordinates implies $p^+ = p^-$, and we keep $p^\varphi$ arbitrary, see (48).

### 4.2.3 Level-matching and mass

We now follow a route analogous to the tensile string [89, 90] and impose the physical state conditions (28) to obtain the physical Hilbert space. For the zero modes, $L_0$ and $M_0$, these give us, respectively, a level-matching condition and a formula for the mass spectrum of the theory. The remaining physical state conditions are automatically satisfied once states are level matched (for a more detailed treatment, see [64]). By virtue of the Oscillator Vacuum (42)-(41), the requirement $L_0|\Psi\rangle = 0$ establishes a level-matching condition

$$s - r = \omega n. \tag{51}$$

From the vanishing of the $M_0$-eigenvalue, $M_0|\Psi\rangle = 0$, we deduce the mass $m := \sqrt{2p^+ p^-}$ of the state $|\Psi\rangle$,

$$m^2 = (r + s)\kappa + \frac{n^2}{R_h^2}. \tag{52}$$

Physical states of a given mass are thus labeled by the integers $r_i, s_i, \omega$, and $n$ subject to the level-matching (51) and the mass-shell condition (52).

# 5 BTZ black hole microstates and their combinatorics

In this section, we introduce and define black hole microstates within the physical Hilbert space of horizon strings discussed in the previous section.

## 5.1 Defining the microstates

We label BTZ black holes by the horizon string mass $m$ and define the set of BTZ black hole microstates as the collection of all physical states in the horizon string Hilbert space. Each microstate

$$|m\rangle_{\text{BTZ}} = |\{r_i\}, \{s_i\}, \omega, n\rangle \tag{53}$$

is labeled by a collection of mode excitation numbers $r_i$, $s_i$, the winding number $\omega$, and the momentum number $n$, subject to the level-matching (51) and the mass-shell condition (52).

The remaining task is to fix the value of the mass $m$ in terms of the geometric input, the value of the horizon radius $R_h$. Since $m$ is the mass of our string states at the horizon, on dimensional grounds it is plausible to identify it (up to some factor) with the near-horizon mass of the associated BTZ black hole. The final puzzle piece is a relation between the BTZ black hole mass and the horizon radius $R_h$, for which we use the near-horizon first law, the essence of which we review.

Assuming the existence of a non-extremal horizon in 3d, in 2015-2016 different boundary conditions have been imposed accounting for the presence of the horizon and fluctuations around it [48, 51, 52, 92, 93]. As explained in [56], this models generic properties of non-extremal horizons in equilibrium with a thermal bath. Our proposal may be viewed as an explicit and microscopic realization of the Hawking, Perry, and Strominger soft hair proposal [49]: the infinite set of near-horizon symmetries generates soft hair excitations. The associated near-horizon charges are conceptually analogous to the asymptotic Brown–Henneaux charges [94]. Like the latter, they obey a first law,

$$\delta E = T \, \delta S \tag{54}$$

where $T$ is the temperature as derived from surface gravity, $S \propto R_h$ is the (Bekenstein–Hawking) entropy, and $E = Q[\partial_t]$ is the near-horizon charge for unit time-translations along the horizon [52]. Integrating the first law (54) requires knowledge of the state-dependence of temperature, e.g., in the form of a Smarr-type relation, a Cardy-type relation, or some assumption about the thermodynamical ensemble. For the near-horizon boundary conditions that lead to soft Heisenberg hair, i.e., to the oscillator algebra (41) as near-horizon symmetries, the temperature is state-independent. Therefore, the first law (54) trivially integrates to

$$E = TS \, . \tag{55}$$

In Cardy language, the entropy depends linearly on the near-horizon energy, in contrast to the asymptotic relation where it scales like the square root of the asymptotic energy.[3]

The near-horizon energy in terms of the near-horizon charges (41) is just the sum of the zero modes $E \propto J_0 + \tilde{J}_0$ (we do not care about the precise numerical coefficient here since below we shall absorb all such factors into a redefinition of our coupling constant $\kappa$), which in turn can be related to the mass of the BTZ black hole as measured by an asymptotic observer, $m \propto J_0 + \tilde{J}_0$ [51, 95].

In summary, due to the near-horizon first law, the mass scales linearly in the horizon radius, i.e.,

$$m = \kappa R_h \, . \tag{56}$$

While we could include some arbitrary numerical factor in the relation (56), we absorb such a factor by the freedom to fix the coupling constant $\kappa$, which we shall do below in (69).

---

[3]The near-horizon version of the Cardy formula can be derived from the anisotropic near-horizon scaling symmetry implicit in near-horizon Rindler-geometries with state-independent surface gravity, see [52].

Plugging the mass (56) into the result (52),

$$\kappa R_h^2 = s + r + \frac{n^2}{\kappa R_h^2} := N + \frac{n^2}{\kappa R_h^2}\,, \tag{57}$$

and assuming $N \gg n$ yields

$$\kappa R_h^2 = N + \frac{n^2}{N} + O(n^4/N^3)\,. \tag{58}$$

In the other limit, $N \ll n$, we have instead $\kappa R_h^2 = n + \frac{N}{2} + O(N^2/n)$.

## 5.2 Combinatorics of microstates

There are various sectors of states, depending on the behavior of the quantum numbers. We discuss all of them and determine their respective combinatorics. In all cases, we assume $R_h \gg 1/\sqrt{\kappa}$ to guarantee the validity of the semiclassical approximation.

### 5.2.1 Soft sector

When the string has vanishing momentum, $n = 0$, we call it soft. In this case, there is a mundane infinite degeneracy from the winding modes: no amount of winding changes anything about the spectrum. We thus declare $n = 0$ states equivalent to each other if they differ only by their winding numbers.

For fixed (large) mass $m$ the counting is now straightforward. The total level $N$ must be large and splits evenly, $s = r = \frac{N}{2}$. The (large) numbers $s$ and $r$ can be partitioned arbitrarily into positive integers. The number of integer partitions, $\Pi(N)$, is given by the Hardy–Ramanujan formula [96] [OEIS: A000041]

$$\Pi(N) \approx \frac{1}{4\sqrt{3}\,N}\, \exp\left(2\pi\,\sqrt{\frac{N}{6}}\right). \tag{59}$$

Thus, the contribution to the partition function from the soft sector (at fixed $m$) is given by

$$Z_{\text{soft}}(N) = \Pi^2\left(\frac{N}{2}\right) \simeq \frac{1}{N^2}\, \exp\left(2\pi\,\sqrt{\frac{N}{3}}\right). \tag{60}$$

Here and in what follows, we use $\simeq$ to denote the approximation of $\ln Z$ to the leading $O(\sqrt{N})$ and subleading $O(\ln N)$ contributions, while dropping terms subleading to these. This simplification has the added benefit that we can assume non-negative $\omega$ and $n$ since considering all possible sign combinations would only produce some overall factor of order unity in the partition function. We make this assumption from now on.

### 5.2.2 High momentum sector

Consider the opposite of the soft sector: The mass is dominated by high momentum, $n \gg N$. Level-matching (51) then implies vanishing winding number $\omega$. So we get the same result as for the soft sector (60), but $N$, while it still can be a large number, is now much smaller than $m^2/\kappa$.

Therefore, this sector is suppressed exponentially as compared to the soft sector and, as we shall see, also as compared to the sectors below. The same logic applies to the sector $n \approx N$. Thus, we conclude that typical microstates require large levels, $N \gg n$, and the contributions to the partition function $Z_{n \gg N} + Z_{n \approx N}$ are negligible for large masses.

### 5.2.3 Generic sector

The higher the level, the stronger the exponential enhancement in the integer partitions (59). Thus, typical microstates require $N \gg n$. Generically, there are no further constraints on winding or momentum other than level-matching and mass-shell conditions. In particular, generically neither of them vanishes, $n > 0$, $\omega > 0$. To reduce clutter, we assume $N$ is even (none of our results change essentially for odd $N$).

A curious aspect of the mass-shell condition (52) is that changing the momentum number $n$ alters the almost-integer number $m^2/\kappa$ slightly. So we should not consider a fixed mass in our ensemble but allow for a range, depending on the allowed range of the momentum number. For the time being, we fix the level $N$ but permit varying the mass by changing the momentum number.

The partition function in the generic sector for fixed $N$

$$Z_{\text{generic}}^{\text{fixed}}(N) = \sum_{l=1}^{\frac{N}{2}} \Pi\left(\tfrac{N}{2} - l\right) \Pi\left(\tfrac{N}{2} + l\right) \tau(2l) \tag{61}$$

involves the number of divisors $\tau(k)$ of the integer $k$. It appears due to the level-matching condition (51), which requires the difference of the levels $s - r$ to be the product $\omega n$. The combinatorial problem (61) is not trivial but solvable at large $N$.

$$Z_{\text{generic}}^{\text{fixed}}(N) \simeq \frac{1}{N^{5/4}} \exp\left(2\pi \sqrt{\frac{N}{3}}\right) \tag{62}$$

We recover the same exponential degeneracy as in the soft sector (60) but with a monomial enhancement in $N$, plus other subleading corrections.

It can be shown that the essential part of the partition function $Z_{\text{generic}}^{\text{fixed}}(N)$ comes from levels $r$ in the range $N/2 - O(N^{3/4})$ to $N/2 - O(1)$, implying typical ranges of winding and momentum numbers between $O(1)$ and $O(N^{3/4})$.

From the range of the momentum number, we deduce an interesting physical fact: Since the momentum number generically scales at most like $O(N^{3/4})$, the quantity $m^2/\kappa$ changes like $O(\sqrt{N})$. We conclude that we must consider Gaussian fluctuations of the level $N$.

We can finally be precise about the allowed mass range in our definition of BTZ microstates. We cannot insist on a fixed value of $m$ but instead must allow fluctuations of the mass, $m \to m + \Delta m$, of order unity to guarantee Gaussian fluctuations in $N$.

$$\Delta m = O(1) \qquad \leftrightarrow \qquad \Delta N = O(\sqrt{N}) \tag{63}$$

It is reassuring that the near-horizon analysis in [51,52] leads to analog conclusions. Namely, if we want Gaussian fluctuations in the BTZ black hole mass as measured by an asymptotic observer, $\Delta M = O(\sqrt{M})$, we need to allow order unity fluctuations of the black hole mass as measured by a near-horizon observer, $\Delta m = O(1)$. It works because $M$ and $m$ are related by a Sugawara construction, yielding $M \propto m^2$. This implies $M + \Delta M \propto (m + \Delta m)^2 = m^2 + m\,O(\Delta m)$, from which we deduce $\Delta M \propto \sqrt{M}\,O(\Delta m)$.

While it was pure combinatorics that drove us to consider Gaussian fluctuations (63), such fluctuations may have been anticipated on physical grounds, as we are in an ensemble of fixed temperature rather than fixed energy.

The number of generic microstates subject to the fluctuations (63) is then given by

$$Z_{\text{generic}}(N) = \sum_{N_0=N}^{N+O(\sqrt{N})} Z_{\text{generic}}^{\text{fixed}}(N_0) \simeq \frac{1}{N^{3/4}} \exp\left(2\pi \sqrt{\frac{N}{3}}\right). \tag{64}$$

### 5.2.4 Non-winding sector

There is one remaining sector, namely $\omega = 0$. In this case, the level-matching (51) implies $s = r$, exactly as in the soft sector. However, since the momentum number $n$ appears in the mass-shell condition (52), we get a cutoff on the spectrum.

For compatibility with the generic sector, we allow the same range of mass fluctuations and hence get Gaussian fluctuations of the level $N$. We obtain

$$Z_{\omega=0}(N) = \sum_{N_0=N}^{N+O(\sqrt{N})} \sum_{n=1}^{O(N_0^{3/4})} \Pi^2\left(\frac{N_0}{2}\right) \simeq \frac{1}{N^{3/4}} \, \exp\left(2\pi \, \sqrt{\frac{N}{3}}\right). \tag{65}$$

While the generic sector contains infinitely more states than the non-winding sector, the approximate equality $Z_{\text{generic}}(N) \simeq Z_{\omega=0}(N)$ shows that the non-winding and the generic sectors yield the same leading and subleading result for the partition function.

## 6  Full partition function and Bekenstein–Hawking law

The final result of our combinatorial excursion is the partition function of BTZ black hole microstates

$$Z_{\text{BTZ}} \simeq Z_{\text{soft}} + Z_{n\gg N} + Z_{n\approx N} + Z_{\text{generic}} + Z_{\omega=0} \,. \tag{66}$$

For large horizon radii, the partition function is dominated by the contribution from the generic sector

$$Z_{\text{BTZ}}(R_h) \approx Z_{\text{generic}}(\kappa R_h^2) \simeq R_h^{-3/2} \, \exp\left(2\pi R_h \, \sqrt{\frac{\kappa}{3}}\right) \tag{67}$$

where we used the relation (58) between the level $N$ and the horizon radius $R_h$. The approximation of the partition function (67) is the main result of our counting. It is valid for large horizon radii, $R_h \gg 1/\sqrt{\kappa}$, which permits comparing with semiclassical results for the black hole entropy.

Summarizing the combinatorics, we find the entropy of our horizon string microstates is given by the logarithm of the partition function (67), viz.,

$$S = \ln Z_{\text{BTZ}} = 2\pi R_h \, \sqrt{\frac{\kappa}{3}} - \frac{3}{2} \, \ln R_h + o(\ln R_h) \,. \tag{68}$$

The entropy (68) contains the correct scaling with the area of the BTZ event horizon, $2\pi R_h$, and a well-known numerical factor $-\frac{3}{2}$ in front of the logarithmic corrections (see, e.g., [67]). The sub-subleading terms are small as compared to $\ln R_h$ but still infinite for $R_h \to \infty$. In particular, they are not of order unity.[4] Comparison with the BH-law (1) fixes the coupling constant as

$$\kappa = \frac{3}{16G^2} \,. \tag{69}$$

The scaling with $1/G^2$ follows from dimensional analysis. The numerical coefficient 3/16 is a non-trivial input and cannot be derived within the setup presented here.

Since the log corrections to the entropy depend on the thermodynamic ensemble let us finally check if we are in the right ensemble. From near-horizon considerations, we have fixed the temperature, and the tensionless string spectrum forced us to let the asymptotic mass have a Gaussian profile, cf. our discussions below (63). The metric (35) in a co-rotating frame has fixed angular momentum. In the conventions of [67], we thus have a mixed ensemble and should get a coefficient of $-\frac{3}{2}$ in front of the log corrections. This is precisely what we obtained in our main result (68).

---

[4]This is qualitatively different from the precision counting e.g. in [97], where the next term is of order unity.

# 7 Concluding remarks

Based on the matching of symmetries of the horizon, we formulated and implemented the idea that microstates of a 3d black hole are physical states (53) of a null string theory. Typical microstates are excited strings with nonzero winding around the angular direction. The growth of their degeneracy with the exponential of the square root of the excitation number (64) is a known characteristic feature of strings [98–100]. This degeneracy correctly accounts for the Bekenstein–Hawking entropy and its subleading log corrections (68).

Before addressing generalizations, we discuss similarities and key differences to some previous proposals for black hole microstates. Here, we compare with three previous proposals: the membrane paradigm [101], fuzzballs [32, 36], and fluffballs [50, 53]; see [102] for more discussions on these three and other proposals. Our construction can be viewed as an implementation of the membrane paradigm [101]. Our mass spectrum (52) is quite different from naive area quantization [37, 103] since, for large black holes, we have a fine (quasi-continuous) spacing between neighboring energy levels, due to the presence of the momentum term $n^2/R_h^2$. Such a fine spacing of energy levels is more in line with general lessons from statistical mechanics than a Planck-quantized area spectrum, see e.g. [104]. Our setup differs conceptually from the fuzzball proposal [32, 36], according to which black hole microstates are horizonless configurations that asymptotically look like a black hole spacetime. By contrast, our construction relied on the existence of a null surface that is identified with the worldsheet of the horizon strings, and the microstates are understood as all possible horizon string configurations, i.e., the microstate degeneracy is due to excitations, winding, and momentum of (tensionless) strings, the ensemble of which effectively (semiclassically) is described by the black hole.

In comparison to the fluff proposal [50, 53], our construction only rests on the near-horizon information, whereas the former used both near-horizon and asymptotic symmetries. Moreover, we avoid ad-hoc input such as the quantization of Newton's constant or the quantization of conical deficit angles that were an integral part of the construction in [50, 53]. The realization of the soft hair proposal in [55] exploits near-horizon Virasoro symmetries and the Cardy formula to account for the black hole entropy but is not explicit about the precise spectrum of microstates.

While our proposal stands firmly on its own, it would be nice to understand how it emerges as a limit of tensile strings approaching a black hole, building on the discussion in [85]. Relatedly, the critical dimension of our horizon strings is 26 [91], and it could be rewarding to consider the impact of these extra dimensions on our analysis. An obvious generalization is to consider higher dimensions, specifically, four spacetime dimensions. Instead of tensionless null strings, one would have to consider tensionless null branes wrapping the horizon. All these issues are works in progress.

# Acknowledgments

It is a pleasure to thank Mangesh Mandlik for helpful correspondence. We thank our colleagues at the mathematics departments at TU Wien and IPM Tehran, in particular Bernhard Gittenberger, Martin Rubey, Iman Eftekhari, and S.M. Hadi Hedayatzadeh, for helpful discussions and proofs in section 5.2 and in particular (62).

AB wishes to acknowledge the support and hospitality of the Erwin-Schrödinger Institute, University of Vienna, and TU Wien (Vienna University of Technology), Austria where the collaboration was initiated. AB was partially supported by a Swarnajayanti Fellowship (SB/SJF/2019-20/08) of the Science and Engineering Research Board (SERB), India, a visiting professorship at École Polytechnique Paris, and by the following grants: CRG/2020/002035 (SERB), Research-in-groups grant (Erwin-Schrödinger Institute). DG was supported by the Austrian Science Fund

(FWF), projects P 30822, P 32581, P 33789, and P 36619 and acknowledges the hospitality of the Perimeter Institute (PI). The final part of this research was conducted while DG was visiting the Okinawa Institute of Science and Technology (OIST) through the Theoretical Sciences Visiting Program (TSVP).

MMShJ is supported in part by SarAmadan grant No ISEF/M/401332 and acknowledges support from the ICTP through the Senior Associates Programme (2022-2027) and Vienna University of Technology, Austria where the collaboration was initiated.

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
