# Peer review of "Horizon Strings as 3d Black Hole Microstates"

_SciPost Physics Core_

## Round 1 · Referee Report · Anonymous · 2023-10-2

Strengths

1. provide a novel understanding of black hole entropy

Weaknesses

1. The physical picture of horizon string is not completely clear

Report

In this article, the authors proposed a novel way to understand the microstates of 3D BTZ black hole. They suggested that the microstates could be described by an ensemble of the string states of horizon string, a kind of null string whose target-space geometry is the black hole horizon. The evidence of the picture is the matching of the microscopic degrees of freedoms with the macro ones. The work could be important, and deserves publication in SciPost.

However, I am not convincing about the coupling constant $\kappa$ introduced first in (2). It is different from the usual string tension, and is fixed to be $3/16G^2$ by matching with the blackhole entropy. The numerical factor was argued to be a nontrivial input, and $1/G^2$ comes from the dimensional analysis. I am confused by the discussion here. In the usual string theory, the string tension is an intrinsic parameter of the theory. Now if we generalize naively the discussion to other dimensions, say 4D Schwartzschild black hole, the dimension of 4D Newton constant is different, and we would expect the coupling constant takes a different form. Is the horizon string picture universal? or we have to adjust the coupling constant case by case? It would be nice for the authors to give more clarification on this point.

Requested changes

The authors missed some references. For example, there's one paper ePrint: 2211.06926 on BMS free fermion, which appeared at the same time with Ref. [82] (ePrint: 2211.06927). Besides, the critical dimensions of tensionless (super)string has been studied from path-integral point of view in e-Print: 2302.05975.

---

## Round 1 · Referee Report · Anonymous · 2023-10-5

Strengths

The manuscript describes 3d Black Hole microstates as arising from quantum states of tensionless strings. The authors consider wound tensionless strings encompassing the black hole horizon and from the degeneracy of the string spectrum calculate the entropy of the black hole and relevant log corrections.

1. The paper is very clearly written with appropriate introduction to the tensionless strings and quantization thereof. The calculation of microstates has also been presented in a lucid way.

2. Both near horizon physics and tensionless worldsheet physics have the inherent structure of BMS symmetries, albeit one is a global symmetry and the other a gauge one. The proposal to put them together is bold and intriguing, however mathematically very well-constructed.

Weaknesses

1. The authors discuss a 3d BTZ Black Hole for their purpose, which could have a non-trivial B field. I do not find a reference to this. As far as I understand the corresponding techniques in Null string theory are also not well developed, so probably this is justified.

Report

I find this paper fully meets the acceptance criteria of the Journal. It is very well-written with impeccable logical flow. The model put forward is completely novel and constructed bottom-up from well known literature. The mathematical realization is rigorous and there are ample scopes of extending this further.

  • validity: top
  • significance: high
  • originality: top
  • clarity: top
  • formatting: excellent
  • grammar: perfect

Author:  Daniel Grumiller  on 2023-10-07  [id 4028]

(in reply to Report 2 on 2023-10-05)
Category:
remark
pointer to related literature

We thank the referee for their comments.

The point about the addition of the B-field is an interesting one. An upcoming paper by one of the authors (with coworkers) will explore null strings with a B-field background. However, as far as our proposal and 3d black hole microstates are concerned, we do not think that the B-field background is relevant.

---

## Editorial Decision

resubmitted